# Analysis of Physical Education Classroom Teaching after Implementation of the Chinese Health Physical Education Curriculum Model: A Video-Based Assessment

**DOI:** 10.3390/bs13030251

**Published:** 2023-03-12

**Authors:** Chao Liu, Cuixiang Dong, Xiaohui Li, Huanhuan Huang, Qiulin Wang

**Affiliations:** 1College of Physical Education, Yangzhou University, Yangzhou 225009, China; 2College of Physical Education and Health, East China Normal University, Shanghai 200241, China; 3School of Foreign Languages, Zhejiang Normal University, Jinhua 311121, China

**Keywords:** Chinese health and physical education curriculum model, teacher behavior, student behavior, interactive behavior, video analysis

## Abstract

This study assessed the Chinese health physical education curriculum model recently suggested to meet the recommended physical education curriculum reforms addressing the declining physical and mental health of students in China. We used video analyses of 41 physical education classroom teaching cases with a physical education classroom teaching behavior analysis system to provide quantitative and qualitative behavioral data. We established reference ranges for classroom teaching behavior indicators, summarized classroom teaching patterns, and assessed classroom discourse and the emotional climate. Notable findings included teachers in elementary schools using closed-ended questions, predictable responses, and general feedback significantly more often than teachers in senior high school, and ball sports instructors using demonstration and competition significantly more frequently than instructors in athletics. Overall, three teaching patterns were most commonly used—lecture, practice, and dialogue—with practice being dominant. Analysis of the top 50 most commonly spoken words by teachers identified five types of discourse—motivational, directive, specialized, transitional, and regulatory—with motivational words being most frequent. The classroom atmosphere was mainly positive. These findings provide evidence that the use of this curriculum model may bring positive changes to physical education classroom teaching methods in China and will inform subsequent innovative physical education classroom teaching practices.

## 1. Introduction

From 1985 to 2010, the General Administration of Sports of China and the Ministry of Education of the People’s Republic of China conducted a total of six national large sample sizes of students’ physical health assessment work, and the results of the study were published. Most indicators show a continuous downward trend. The detection rates of obesity and overweight among students continue to increase, with the detection rates of overweight among urban boys, urban girls, rural boys, and rural girls aged 7–22 years old are 14.81%, 9.92%, 10.79% and 8.03%, respectively, an increase of 1.56, 1.20, 2.59, and 3.42 percentage points, respectively, compared with 2005 [1]. According to the definition of the International Obesity Working Group, there are 12 million overweight and obese children and adolescents in China, and among the 155 million overweight and obese children and adolescents in the world, one out of every 13 is a Chinese child or adolescent [2]. In terms of mental health, the Chinese Association for Science and Technology, together with the Chinese Psychological Association, has conducted a study on the “Survey on the Mental Health of Chinese Adolescents”, the results of which show that the proportion of adolescents with serious psychological problems in China is high, with 17.5% of adolescents having psychological problems and 3.1% having serious psychological problems; 63.3% of adolescents were depressed, 29.1% were often nervous and upset, and 31.7% had more feelings of anger [3]. Using a cross-sectional historical study, Xin et al. examined changes in the mental health of secondary school students in China since 1992 and found that the overall level of mental health of secondary school students has gradually decreased and that the differences in mental health levels among secondary school students have now increased, meaning that many secondary school students may currently have serious psychological problems [4]. Therefore, in response to the declining physical and mental health of students in China, the Chinese health physical education curriculum model was developed. Any kind of physical education curriculum model is based on solving the main problems, and the Chinese health physical education curriculum model is no exception. For example, the physical education curriculum model focuses on how to improve students’ physical fitness [5]; the sports education curriculum model focuses on improving students’ ability to compete [6]; the personal and social responsibility curriculum model focuses on cultivating students’ integrity and self-discipline, compliance with rules, respect for others, willingness to help others, and communication and cooperation through sports [7]; and the Chinese health physical education curriculum model focuses on solving the physical and mental health of students. The curriculum model combined local characteristics with an international perspective and included the basic concepts outlined in the national Physical Education and Health Curriculum Standards to enable students to achieve healthy physical and mental development [8]. The curriculum model puts forward clear overall requirements for classroom teaching, including that teachers should implement the principle of “more practice and less lectures”, create real, complex learning and activity situations, and integrate modern information technology tools with classroom teaching. Through smart devices such as wearable sensors, holographic camera systems, infrared imaging, and eye tracking, multi-modal data in the student ‘s learning process are dynamically collected and managed, such as speech data, behavioral data, psychological data, physiological data, and brain data. Multi-modal data interact to fully describe the students’ learning process, establish a big data platform for primary and secondary school students’ sports and health, create a full-life cycle health guarantee system, and achieve a comprehensive quantification of the learning process. Students should be encouraged to use an independent but cooperative inquiry learning style, with teachers guiding students to learn and practice structured sports and skills while allowing them independent and active learning time and space. Teachers should strive to move away from a teaching-oriented method, toward a learning-oriented method, and create a flexible, diverse, and dynamic learning environment in the classroom. Teachers should encourage harmonious interactions between themselves and the students, as well as among students, and establish a positive classroom teaching atmosphere. The assessments of motivation, feedback, guidance, development, etc., should consider both teacher and student appraisals. The curriculum model is the first in the field of physical education in China complying with the Physical Education and Health Curriculum Standards. The concept and method are novel, advanced, and highly operational. As an increasing number of teachers are adopting this curriculum model, students are benefiting [9].

The new round of physical education curriculum reform in basic education in China began in 2001, and as of today, it has gone through a twenty year journey. If we take 10 years as the interval and summarize the achievements and experiences in the past 20 years, the focus of the physical education curriculum reform in basic education in the first 10 years is the change of the curriculum idea and teaching concept, while the focus of the physical education curriculum reform in basic education in the last 10 years is the improvement of the quality of the physical education classroom teaching. Therefore, by analyzing the physical education classroom teaching using this curriculum model, we can evaluate the effectiveness of this reform in China and provide information for further physical education curriculum reform. As one of the important indicators to improve and evaluate the quality of classroom teaching, physical education classroom teaching behavior has become an important entry point for this study. Physical education classroom teaching behavior refers to the sum of various behavioral activities between teachers and students to accomplish specific teaching objectives in a specific physical education classroom context under the guidance of certain physical education teaching ideas, which specifically includes teacher behavior, student behavior, and interactive behavior. These three behaviors interact and influence each other, and as an organic whole, together determine the teaching quality of a physical education class.

In the field of educational research, video analysis methods began to emerge in in the 1970s. In 1974, the National Institute of Education held a subforum entitled “Teaching in Cultural Contexts: As Linguistic Processes”, which attracted widespread academic attention [10]. In 1977, that institute funded 10 research projects that explicitly supported the exploration of educational issues through the documentation of classroom processes. Most of the leaders of these 10 research projects used video analysis, and they became the founders of the application of video analysis to the field of educational research [11]. After more than 50 years of development, video-based empirical research has become a mainstream approach to study the learning processes in physical education classrooms. Video analysis has been widely used to advance many areas of educational research, including classroom teaching behavior [12], classroom teaching patterns [13], class-room discourse [14], and the classroom climate [15].

In this study, we used a mixed method of qualitative and quantitative approaches and applied physical education classroom teaching behavior as an analysis tool, aiming to address the real state of physical education classroom teaching implementation under the Chinese health physical education curriculum model. We attempted to establish a reference range of classroom teaching behavior indicators, summarize classroom teaching patterns, and assess classroom discourse and emotional climate to inform future innovative physical education classroom practices.

## 2. Materials and Methods

### 2.1. Participants and Procedure

From the beginning of 2016, the research team of Chinese health physical education curriculum model gradually set up experimental bases in more than 50 primary and secondary schools in several provinces and cities across the country, and more than 100,000 students from base schools participated in the experiment. The vast majority of physical education teachers in the base school have participated in different forms of centralized training, and the members of the research team often go to the base school to carry out guidance and communication. Every year, a classroom teaching demonstration exchange activity is held between the base schools. In 2016–2018, the National School Sports Alliance (physical education) also held a classroom teaching demonstration exchange activity for the Chinese health physical education curriculum model for three consecutive years. All on-site classroom teaching demonstrators were selected from the base. High-quality lessons have attracted thousands of physical education teachers, experts, and scholars from all over the country to observe and communicate, and have received extensive media attention and reports [16,17,18]. Therefore, files of high-quality video obtained in classrooms, in which teachers were using the Chinese health physical education curriculum model at the 4th, 5th, and 6th National School Sports Federation (Physical Education) Congress, were selected for this study. The learning environment and settings are shown in Figure 1 and Figure 2. The videos included 41 lessons taught across 13 elementary schools, 10 junior high schools, and 18 senior high schools. Each session lasted for approximately 45 min. On the day of filming, the researcher arrived at the gym at least 5 min before the scheduled start time. A small clip-on microphone and recorder were attached to the instructor. The researcher remained in an inconspicuous location away from the classroom instruction and recorded the session until all students had completed their studies and left the area. To prevent a possible Hawthorne effect [19], the researcher visited the teacher in the classroom in advance of the filming. This ensured that the teacher and students were familiar with the researcher’s presence at the time of filming.

### 2.2. Measures

We used the classroom teaching behavior analysis system associated with the Chinese health physical education curriculum model [20]. The analysis consisted of three parts: a coding system, criteria for observing and recording codes, and methods for recording and analyzing behaviors. The coding system consisted of three dimensions: teacher behavior, student behavior, and interactive behavior. Teacher behavior was divided into four categories, explanation and demonstration, instruction and evaluation, information technology application, and transition, coded as 1–4, respectively. Student behavior was divided into 4 behavior categories, preparation activity practice, motor skill practice, physical fitness practice, and relaxation activity practice, coded as 5–8, respectively. Student–teacher interactive behavior was divided into 4 categories, questioning and responding, cooperating with learning and training, teacher-student co-competition, and mutual discussion and evaluation, coded as 9–12, respectively (Table 1).

The analysis system used both quantitative and qualitative recording methods. For the quantitative recording, a temporal sampling method was used to collect a sample of behaviors at 10-s intervals. The behaviors were coded based on the behavior categories. For the qualitative recording, a combination of technical records and narrative systems was used.

The systematic analysis methods included cluster analysis, information entropy analysis, redundancy analysis, ratio analysis, verbatim analysis, and temporal analysis, each of which achieved different purposes in physical education classroom analyses (Figure 3). Cluster analysis reflected the distribution of teaching patterns. Information entropy and redundancy analysis reflected the amount of teaching information. Ratio analysis reflected the structure within the classroom. Verbatim analysis can reflect the discourse and emotional climate. Temporal analysis can reflect the process of teaching changes.

To ensure the reliability and validity of the study, three trained observers independently coded 10 identical videotapes of physical education classes to assess inter-rater reliability. One observer coded the same 10 classes twice, 2 weeks apart, to assess intra-rater reliability. Inter- and intra-observer agreement (reliability) of the three coders’ ratings was examined using a two-way random intra-class correlation coefficient (ICC) [21]. The results showed that inter-observer (MICC = 0.94 to 0.97) and intra-observer (MICC = 0.96) reliability reached very good levels. In addition, the results showed that students’ learning behaviors were influenced not only by their instructors’ teaching behaviors (t = 3.52, *p* = 0.00) but also by teaching interactions (t = 2.26, *p* = 0.03), indicating that the classroom teaching behavior analysis system had good predictive validity [22].

### 2.3. Statistical Analysis

The reference ranges for each dimension of classroom teaching behavior were determined using the non-parametric Kruskal–Wallis test for school stage (elementary, junior high, and senior high schools) and sport (ball sports, gymnastics, and athletics), and the non-parametric Mann–Whitney test for teacher gender and post hoc multiple comparisons of the differences in the distribution of each dimension of teaching behavior. Shapiro–Wilk tests were used to determine whether the data were normally distributed for each dimension of teaching behavior. The final determination of the reference range for each dimension of teaching behavior was based on statistical outcomes, as well as the concepts and requirements in the Chinese health physical education curriculum model. Classroom teaching patterns were assessed using K-means cluster analysis for the frequency data in the three dimensions of teacher behavior, student behavior, and interactive behavior. Discourse word frequency analysis and emotional function analysis were conducted using Nvivo, Version 12, to analyze the classroom discourse and emotional climate, respectively.

## 3. Results

### 3.1. Establishment of a Reference Range for Each Indicator of Classroom Teaching Behavior

We established a reference range for each indicator of classroom teaching behavior implemented through the Chinese health physical education curriculum model to enable the visualization of the current teaching structure and to provide a practical reference for subsequent physical education classroom teaching.

#### 3.1.1. Differences in Classroom Teaching Behaviors by Student School Stage, Sport, and Gender

Non-parametric tests were conducted with school stage (elementary, junior high, and senior high school), sport (ball sports, gymnastics, and athletics), and teacher gender (male and female) as independent variables, and each indicator of classroom teaching behavior as a dependent variable. The results are given in Table 2.

The frequencies of closed-ended questioning, predictable response, and general feedback showed statistically significant differences by school stage (*p* = 0.02). The frequencies of display and competition (*p* = 0.02) showed statistically significant differences by sport. However, none of the indicators of classroom teaching behavior showed statistically significant differences by teacher gender.

Further post hoc multiple comparisons indicated that closed-ended questioning, predictable response and general feedback were significantly different between elementary school and senior high school (*p <* 0.001), with elementary school (median, 13.00; inter-quartile range, 10.00–15.00) greater than senior high school (median, 6.00; inter-quartile range, 2.00–10.00). Display and competition were significantly different between ball sports and athletics (*p* = 0.01), with ball sports (median, 42.00; inter-quartile range, 36.00–47.00) being greater than athletics (median 20.5; inter-quartile range, 9.80–34.30). On the basis of these findings, the reference ranges were established for closed-ended questioning, predictable response, and general feedback at the different school stages, the reference ranges were established for the indicators of display and competition at different sports and universal reference ranges were established for other indicator of classroom teaching behavior.

#### 3.1.2. The Specific Numerical Value Establishment of the Reference Range of Each Indicator of Classroom Teaching Behavior

The data for each indicator of teacher behavior, student behavior, and interactive behavior under the Chinese health physical education curriculum model were tested for normality using the Shapiro–Wilk test (Table 3). Data for the following indicators were not normally distributed (i.e., *p* < 0.05): instruction and evaluation, information technology application, transition, preparation activity practice, individual technique practice, physical fitness practice, open-ended questioning, comprehensible response, professional feedback, cooperating with learning and training, teacher–student co-competition, and mutual discussion and evaluation. By contrast, data for the following indicators were normally distributed (i.e., *p* > 0.05): explanation and demonstration, combined technique practice, display and competition, relaxation activity practice, closed-ended questioning, predictable response, general feedback, teacher behavior, student behavior, interactive behavior, and total time.

The reference ranges were determined by selecting the bilateral limits (i.e., mean ± (1.96 × SD)) or the one-sided upper limit (i.e., mean + (1.645 × SD)) and the one-sided lower limit (i.e., mean − (1.645 × SD)) within the 95% confidence interval for each indicator that was normally distributed. Reference ranges for indicators that were not normally distributed were established using the non-parametric percentile method with double-sided limits (2.5%, 97.5%) or with a single-sided upper limit (95%) or a single-sided lower limit (5%) (Table 4).

### 3.2. Analysis of Classroom Teaching Patterns

The teaching patterns used in the physical education classrooms implementing this model were assessed with K-means cluster analysis using the frequency data from the three dimensions—teacher behavior, student behavior, and interactive behavior—as the base. The results indicated that it was reasonable to divide the classroom teaching patterns into three types. The descriptive statistics for these three types are given in Table 5. The means for each dimension of physical education classroom teaching behavior were significantly different for the three different teaching patterns (teacher behavior, F = 20.17, *p* < 0.01; student behavior, F = 30.27, *p* < 0.01; and interactive behavior, F = 8.95, *p* < 0.01). The percentages of the three physical education classroom teaching patterns were 32.26%, 29.03%, and 38.71%, which were evenly distributed, indicating a good clustering effect.

The first type of physical education classroom teaching pattern was most prominent for the dimension of interactive behavior, with teacher behavior being more prominent and student behavior less so. Taken together with the overall requirements of the Chinese health physical education curriculum model for physical education classroom teaching, this type of classroom can be summarized as a dialogue type and accounts for 32.26% of the total. In this type of classroom, teachers and students are in dialogue with each other. The role of the teacher is to make decisions about the content of the physical education, including the target concepts to be discovered and the sequence of questions to be set for the students. The students’ role is to reason, ask questions, and discover accurate answers during the learning and practice processes.

The second teaching pattern was most prominent for the dimension of teacher behavior, with student behavior being more prominent and interactive behavior being less prominent. This type of physical education classroom can be summarized as a lecture II type and accounts for 29.03% of the total. Compared with a traditional classroom in which the physical education teacher explains and demonstrates (lecture I), a lecture-II-type classroom has time for explanations and demonstrations by the teacher but also leaves time for students to practice.

The third type of physical education classroom teaching pattern was most prominent in the student behavior dimension, interactive behavior was more prominent, and teacher behavior was not prominent. This type of classroom can be summarized as a practice type and accounts for 38.71% of the total. In this type of classroom, the role of the physical education teacher is to make decisions about the content and organization, and to provide timely feedback to the students, who spend most of their time engaging in individual, group, and collective practice and communication.

### 3.3. Analyses of Classroom Discourse and Emotional Climate

#### 3.3.1. Discourse Analysis

In order to analyze the discourse characteristics of physical education teachers, the top 50 most commonly spoken words were selected by importing the transcribed classroom teaching audio text into Nvivo 12 for discourse word frequency analysis. The results of the analysis are shown in Figure 4.

The discourse of physical education teachers was divided into five major categories based on word frequency and presented herein in descending order.

The first category was motivational words, for example, cheer, can, and great. Teaching under this model focuses on the motivational and developmental functions of assessment, which requires teachers to give more praise or encouragement to students. Appropriate praise or encouragement can stimulate student motivation, creativity, and desire to perform, so that students have a successful sports experience, which may aid in achieving a healthy personality and good character development.

The second category was directive speech, for example, ready, start, and pay attention. Instructions are key initiatives in organizing and sustaining physical education classroom instructions and having a positive effect on classroom management. The appropriate use of directive speech by teachers may trigger or mobilize students to participate in the classroom activities, stimulate student interest in learning, inspire student thinking, enhance interactions between teachers and students, and improve the effectiveness of classroom teaching.

The third category of words used was specialized speech, for example, passing, defense, and coverage. Specialized speech mainly plays a guiding role. Teachers give reinforcement and feedback to students who are practicing in response to the actual situation, and promote the formation of correct action awareness. Teachers may either stop students for focused instruction or give “tour guidance” as students continue to practice.

The fourth category was transitional speech, for example, words such as then, next, and last. Transitional speech mainly serves as a link between physical activity and physical education content. In the classroom, the appropriate use of transitional speech effectively reduces the time it takes to teach team movement, field layout, pauses, and the like, increases the time for students to learn and practice motor skills, and improves the logic and hierarchy of classroom teaching.

The fifth category was regulatory speech, with words such as breathing, adjustment, and rhythm. In the physical education classroom, the correct use of regulatory speech helps students stay within a reasonable exercise load, promotes the development and recovery of their physical functions, avoids sports injuries, and ensures their health and safe participation in exercise.

#### 3.3.2. Emotional Climate Analysis

Emotional climate was assessed using automatic coding with Nvivo 12 to identify emotional functions (Figure 5).

There were 235 very negative reference points, 224 more negative reference nodes, 1598 more positive reference nodes, and 280 very positive reference nodes. The sum of the more positive and very positive nodes (1878) was substantially greater than the sum of the more negative and very negative nodes (459), indicating that the emotional climate of the physical education classroom under this model was predominantly positive, and the classroom atmosphere was very good.

## 4. Discussion

This study established reference ranges by using statistical methods consistent with existing related studies for: (1) Closed-ended questioning, predictable response, and general feedback at the level of elementary, junior high, and senior high school; (2) Indicators of different project displays and competitions; and (3) Each indicator of other teaching behaviors. For example, one previous study used the percentile method to determine bilateral reference ranges for blood and serum iron levels in Tibetan children aged 2–14 years based on the distribution characteristics of test indicators [23]. Another study used the percentile method to determine bilateral 95% limits and to establish reference ranges for anemia-related testing indexes in older adults and centenarians in Hainan, China [24]. In the present study, the use of closed-ended questioning, predictable response, and general feedback in physical education classes implementing the Chinese health physical education curriculum model was significantly different between elementary school and senior high school (*p* < 0.01), with greater usage in elementary school than in high school. This is consistent, to some extent, with the findings of previous studies. One study found that at the elementary level, teachers spend the vast majority of their classroom time engaged in the question–answer–feedback cycle, with teachers asking an average of more than 100 questions per day [25]. Classroom questioning is a key tool for promoting the development of student thinking skills. Especially at the elementary level, the content, format, and number of questions teachers ask play a critical role in the development of the students’ mind [26].

The present study also found that the use of display and competition in classrooms implementing the curriculum model was significantly different for ball sports and athletics (*p* < 0.05), with greater usage in ball sports than in athletics. Plausible reasons for this finding are that athletics involve closed motor skills with periodic and procedural characteristics. The key to the formation of motor skills lies in repeated practice until a standard pattern and degree of automation is achieved. By contrast, ball sports, such as basketball, involve open motor skills with confrontational and open characteristics. The formation of motor skills in ball sports requires both static and repeated reinforcement practice of technical movements and the rational use of technical movements in dynamic game situations. It should be noted that this explanation is based on assumptions and generalizations about the nature of the motor skills required for each sport, which may not be appropriate for all people or situations. There may be other factors that influence display and competition in different sports, such as cultural norms, teacher preferences, and student interests.

Our K-means cluster analysis indicated that three teaching patterns were predominately used under the Chinese health physical education curriculum model—dialogue, lecture II, and practice, with the highest percentage of classrooms using the practice type. This finding indicated a positive change in teaching patterns. This gives empirical support to previous research to a certain extent; for example, Liu reported that ensuring the athletic load of physical education classes meant that it would be necessary to change the traditional physical education teaching styles away from the “didactic”, “technical”, “safety”, and “military” forms that have been used for decades and build new forms that meet the spirit and requirements of curriculum standards [27]. All three teaching styles used most with the curriculum model helped to achieve the goal of cultivating student core literacy in physical education. The form teachers ultimately choose to follow will depend on the learning situation, the characteristics of the classroom materials, and the amount of class time allotted. For example, Griffey used command and task-based teaching styles to teach volleyball forearm passing and serving skills. The results showed that high school students with higher skill levels performed better in task-based teaching.; Jenkins and Byra used the practice style to teach school-aged children to hit the ball with a racket. The results showed that using the practice style to teach can effectively promote school-aged children’s motor skills learning [28].

Using Nvivo 12 word frequency analysis, the present study found five main discourse systems used by teachers instructing under the Chinese health physical education curriculum model: motivational speech, directive speech, specialized speech, transitional speech, and regulatory speech. This finding provides empirical support to the results of some previous studies. For example, one study classified classroom discourse into three categories from the perspective of the function of the discourse: course content discourse (specialized speech and regulatory speech), discourse of social control in the classroom (directive speech and transitional speech), and discourse that expresses the individual’s personality (motivational speech) [29]. Classroom discourse is an important factor affecting teaching effectiveness, and it is also an important reflection of the quality of a physical education teachers’ speech and discourse efficiency [30]. The appropriate use of discourse not only promotes the development of student thinking, but also stimulates students’ emotions and has an emotional impact on them. Some studies have shown that through a teacher’s encouraging attitude and appreciative language, students gain psychological comfort, develop a better understanding and knowledge of the learned content, and become more motivated to participate in classroom questions and activities. This type of discourse helps create a learner-centered teaching atmosphere that better facilitates teaching and learning [31].

The present study also found that classrooms using the Chinese health physical education curriculum model had a predominantly positive emotional climate with a good atmosphere. A positive classroom climate helps to motivate and initiate student learning, assists in exploring students’ learning potential, and encourages students to enjoy sports [32]. This finding provides empirical support to previous study findings. For example Liu suggested that classroom teaching under the Chinese health physical education curriculum model should focus on creating a classroom teaching atmosphere with harmonious teacher–student interactions, positive emotions, lively and enthusiastic scenarios, and a positive atmosphere. A good classroom teaching atmosphere plays a direct role in influencing students’ learning abilities and effectiveness, enables active learning, and allows students to experience the joy of learning [33].

## 5. Conclusions

In summary, this study established the frequency and time reference ranges for closed-ended questioning, predictable response, general feedback at three different school stages, for display and competition indicators at different sports, and for other teaching behavior indicators. Of the three classroom teaching patterns (dialogue, lecture II, and practice) used most often under the Chinese health physical education curriculum model, practice was predominant. Teachers instructing under this curriculum model used five types of discourse (motivational speech, directive speech, specialized speech, transitional speech, and regulatory speech), and the emotional climate in the physical education classroom was mainly positive.

The video case analysis tool in this study can be used as a reference for other researchers to conduct related studies. By analyzing the reference range of each indicator of physical education classroom teaching behavior, generalized physical education classroom teaching patterns, classroom discourse, and emotional climate, not only provides a more systematic and in-depth understanding of current physical education classroom teaching, but also can be used to improve the quality of physical education teaching in China.

This study has limitations that should be considered. Owing to the limitations of the video samples used in this study, our findings and conclusions may not be generalizable to all physical education programs in China. Thus, future studies should increase the sample size to collect more data to strengthen the generalizability of the study findings.

## Figures and Tables

**Figure 1 behavsci-13-00251-f001:**
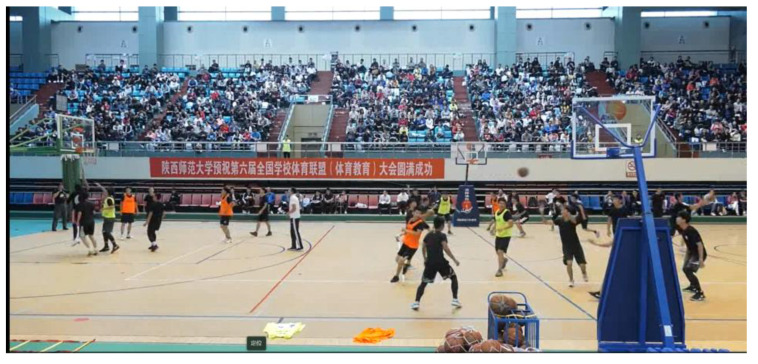
Senior high schools’ learning environment and settings.

**Figure 2 behavsci-13-00251-f002:**
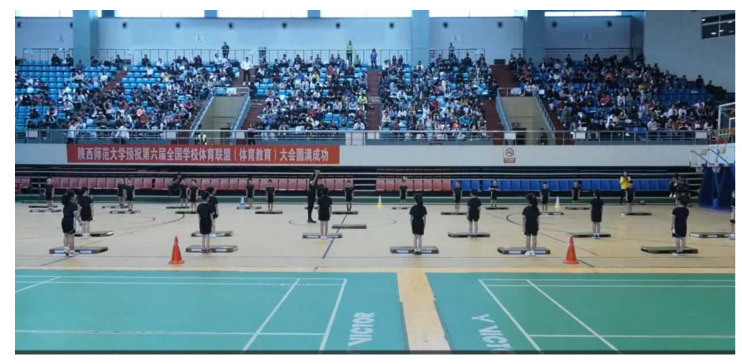
Elementary schools’ learning environment and settings.

**Figure 3 behavsci-13-00251-f003:**
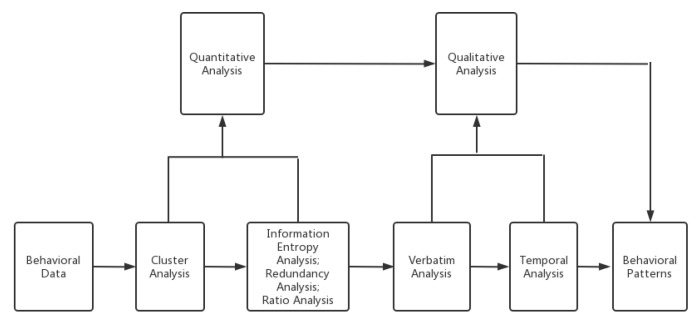
Schematic diagram depicting the quantitative and qualitative teaching behavior analysis methods used in the study.

**Figure 4 behavsci-13-00251-f004:**
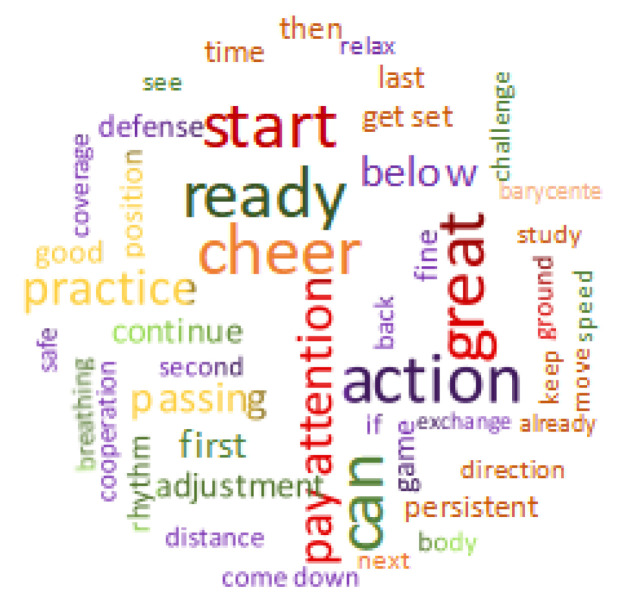
Top 50 words spoken by physical education teachers in sample physical education classes. Note: the size of words in the schematic diagram, from small to large, represents the frequency of use, from less to more, respectively; color represents the category of discourse.

**Figure 5 behavsci-13-00251-f005:**
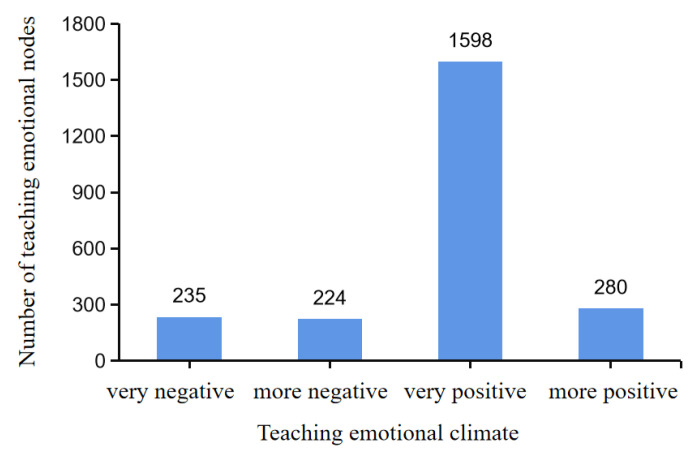
Example reference nodes for teaching emotional climate.

**Table 1 behavsci-13-00251-t001:** Behavior coding system.

Classification	Code	Behavior	
Teacher behavior	1	Explanation and demonstration	
	2	Instruction and evaluation	
	3	Information technology application	
	4	Transition	
Student behavior	5	Preparation activity practice	
	6 Motor skill practice		6.1 Individual technique
			6.2 Combination technique
			6.3 Display and competition
	7	Physical fitness practice	
	8	Relaxation activity practice	
Interactive behavior	9 Questioning and responding		9.1 Closed-ended questioning, predictable response, general feedback 9.2 Open questioning, understanding response, professional feedback
			9.3 Open-ended questioning, comprehensible response, professional feedback
	10	Cooperating with learning and training	
	11	Teacher–student co-competition	
	12	Mutual discussion and evaluation	

**Table 2 behavsci-13-00251-t002:** Non-parametric analysis of each indicator of classroom teaching behavior, by school stage, sport, and student gender.

Indicator	School Stage		Sport		Student Gender	
	Kruskal–Wallis H Statistic	*p*	Kruskal–Wallis H Statistic	*p*	Mann–Whitney z Score	*p*
Explanation and demonstration	0.49	0.78	1.56	0.46	−0.26	0.79
Instruction and evaluation	2.63	0.27	3.70	0.16	−0.31	0.76
Information technology application	0.19	0.91	2.71	0.26	−0.51	0.61
Transition	1.19	0.55	2.19	0.34	−0.15	0.88
Preparation activity practice	1.06	0.59	3.17	0.21	−0.89	0.37
Individual technique practice	1.45	0.49	0.23	0.89	−0.43	0.66
Combination technique practice	0.09	0.96	5.27	0.07	−1.63	0.10
Display and competition	2.62	0.27	8.05	0.02 *	−0.22	0.83
Physical fitness practice	0.32	0.85	0.17	0.92	−0.61	0.54
Relaxation activity practice	1.97	0.37	0.48	0.79	−0.13	0.90
Closed-ended questioning, predictable response, general feedback	7.52	0.02 *	3.70	0.16	−0.31	0.76
Open-ended questioning, comprehensible response, professional feedback	2.63	0.27	0.88	0.65	−1.49	0.14
Cooperating with learning and training	4.19	0.12	0.90	0.64	−0.83	0.41
Teacher–student co-competition	3.51	0.17	6.00	0.09	−0.74	0.46
Mutual discussion and evaluation	0.98	0.61	4.34	0.11	−1.45	0.15
Teacher behavior (total)	5.35	0.07	1.70	0.43	−0.52	0.60
Student behavior (total)	0.65	0.72	4.75	0.09	−0.39	0.70
Interactive behavior (total)	0.61	0.74	0.32	0.85	−1.79	0.07

Note: * significant at the 0.05 level.

**Table 3 behavsci-13-00251-t003:** Normality test results for each indicator of physical education classroom teaching behavior.

Indicator	Frequency Mean ± SD	Time (min) Mean ± SD	Shapiro–Wilk *p* Value
Explanation and demonstration	25.76 ± 8.89	4.29 ± 1.48	0.96
Instruction and evaluation	11.59 ± 5.07	1.93 ± 0.85	0.92 *
Information technology application	0.93 ± 2.78	0.16 ± 0.46	0.38 **
Transition	19.00 ± 7.33	3.17 ± 1.22	0.93 *
Preparation activity practice	31.24 ± 8.52	5.21 ± 1.42	0.89 **
Individual technique practice	13.79 ± 17.30	2.30 ± 2.88	0.80 **
Combination technique practice	44.52 ± 19.59	7.42 ± 3.26	0.93
Display and competition	30.66 ± 16.89	5.11 ± 2.81	0.96
Physical fitness practice	38.21 ± 8.83	6.37 ± 1.47	0.91 *
Relaxation activity practice	12.97 ± 4.23	2.16 ± 0.70	0.94
Closed-ended questioning, predictable response, general feedback	8.10 ± 4.64	1.35 ± 0.77	0.94
Open-ended questioning, comprehensible response, professional feedback	1.83 ± 1.83	0.30 ± 0.31	0.86 **
Cooperating with learning and training	6.59 ± 5.17	1.10 ± 0.86	0.93 *
Teacher–student co-competition	3.86 ± 6.11	0.64 ± 1.02	0.69 **
Mutual discussion and evaluation	5.07 ± 4.44	0.85 ± 0.74	0.91 *
Teacher behavior	57.79 ± 14.10	9.63 ± 2.35	0.93
Student behavior	173.93 ± 18.22	28.99 ± 3.04	0.94
Interactive behavior	26.62 ± 9.97	4.44 ± 1.66	0.95
Total time		43.16 ± 2.76	0.95

Note: * significant at the 0.05 level. ** significant at the 0.01 level.

**Table 4 behavsci-13-00251-t004:** Reference ranges for each classroom teaching behavior indicator.

Indicator	Reference Range (Frequency, No. of Observations)		Reference Range (Min)
Explanation and demonstration	(10.00, 43.00)		(1.39, 7.19)
Instruction and evaluation	(5.00, 21.00)		(0.83, 3.50)
Information technology application	(>0.00)		(>0.00)
Transition	(8.00, 36.00)		(1.33, 6.00)
Preparation activity practice	(22.00, 52.00)		(3.66, 8.67)
Individual technique practice	(<51.00)		(<8.50)
Combination technique practice	(45.00, 77.00)		(7.42, 12.80)
Display and competition	ball sports	(44.00, 60.00)	(7.33, 10.00)
	athletics	(38.00, 52.00)	(6.33, 8.67)
	gymnastics	(24.00, 42.00)	(4.00, 7.00)
Physical fitness practice	(19.00, 50.00)		(3.17, 8.33)
Relaxation activity practice	(5.00, 21.00)		(0.79, 3.53)
Closed-ended questioning, predictable response, and general feedback	elementary school	(<18.00)	(<3.00)
	junior high school	(<16.00)	(<2.67)
	senior high school	(<14.00)	(<2.33)
Open-ended questioning, comprehensible response, professional feedback	(>0.00)		(>0.00)
Cooperating with learning and training	(0.00, 21.00)		(0.00, 3.50)
Teacher–student co-competition	(0.00, 20.00)		(0.00, 3.33)
Mutual discussion and evaluation	(0.00, 16.00)		(0.00, 2.67)
Teacher behavior	(30.00, 85.00)		(5.02, 14.24)
Student behavior	(138.00, 210.00)		(24.17, 34.50)
Interactive behavior	(7.00, 46.00)		(1.19, 7.69)
Total time			(>38.61)

**Table 5 behavsci-13-00251-t005:** Descriptive statistics for the three types of physical education classroom teaching patterns in each of three dimensions.

Dimension	Type 1 (Mean ± SD)	Type 2 (Mean ± SD)	Type 3 (Mean ± SD)	F/P
Teacher behavior	65.58 ± 14.37	67.78 ± 9.43	40.80 ± 4.02	20.17 **
Student behavior	154.33 ± 8.57	183.67 ± 14.51	189.60 ± 11.39	30.27 **
Interactive behavior	33.92 ± 12.44	16.89 ± 6.31	27.60 ± 5.97	8.95 **
Percentage	32.26%	29.03%	38.71%	

Note: ** significant at the 0.01 level.

## Data Availability

All data generated or analyzed during this study are included in this published article.

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
