# Peer review of "Analysis of Physical Education Classroom Teaching after Implementation of the Chinese Health Physical Education Curriculum Model: A Video-Based Assessment"

_behavsci, 2023, doi:10.3390/bs13030251_

Round 1

Reviewer 1 Report

Line 37 it phys-ical or physical? if it is phys-ical what means this separation?

Have many words like that cur-riculum, develop-ment, require-ments, and so on. I think it is problem of edition. 

You are speaking about curriculum reform, I believe it is necessary to explain how was before and how it is now, what are the principal differences. 

After line 58 don't have connection with the previous development.  

In the introduction you must clarified research design, and research question. Because it is no clear what do you want to know... behaviors related to what?

How the three kind of behavior are connected? teacher behavior, student behavior and interactive behavior? how they have influence between there?

In findings you do reference to predictable response, what it means? what are the consequence of this kind of response?

It can be interesting if your findings dialogue with some literature, and give more meaning to this. 

Reviewer 2 Report

Thank you for the opportunity to read this manuscript. The study seems to have employed a rigorous research methodology, including the use of a physical education classroom teaching behavior analysis system, video analyses of 41 physical education classroom teaching cases, and the establishment of reference ranges for classroom teaching behavior indicators. The study's findings regarding the use of different teaching patterns and types of discourse in physical education classrooms in China also appear to be of potential interest to educators and researchers in the field. I have several comments to help improve the readability of the paper:

1.       The authors should provide more background information on the declining physical and mental health of students in China, such as statistics or research findings, to emphasize the importance of the curriculum model being developed.

2.       Their introduction section should clarify the specific characteristics of the Chinese Health Physical Education Curriculum Model, and how it differs from previous physical education curriculums in China, to help readers understand the context and significance of the study.

3.       They could also elaborate on how the curriculum model integrates modern information technology tools with classroom teaching and why this is important for promoting healthy physical and mental development.

4.       The authors should provide more information on the criteria used for selecting the schools and classrooms included in the study. Providing this information would help readers understand the sampling process and the representativeness of the sample.

5.       The title suggests the use of a video-based assessment: Would the authors be willing to add a few screenshots from the experiments (blurring personal information, if any) to help readers clearly visualize the learning environment and settings?

6.       The authors presented a plausible explanation for why the use of demonstration and competition is significantly different in ball sports compared to athletics in classrooms implementing the curriculum model. However, their explanation is based on assumptions and generalizations about the nature of motor skills required in each sport, which may not be accurate for all individuals or situations. 

7.       Additionally, I wonder where the author could consider other factors that may affect the use of demonstration and competition in different sports, such as cultural norms, teacher preferences, and student interest.

8.       While the conclusion mentions the indicators of physical education classroom teaching behavior that were established in the study, it does not provide a clear summary of the main findings. A brief summary of the key results should be included to ensure that readers understand the significance of the study.

9.       The conclusion briefly mentions that the study provides practical guidance for teachers who want to use innovative techniques, but this point could be expanded upon. Specifically, the conclusion should emphasize how the established reference ranges and teaching behavior indicators can be used to improve the quality of physical education instruction in China.
